# Portable Medical Suction and Aspirator Devices: Are the Design and Performance Standards Relevant?

**DOI:** 10.3390/s22072515

**Published:** 2022-03-25

**Authors:** Saketh R. Peri, Forhad Akhter, Robert A. De Lorenzo, R. Lyle Hood

**Affiliations:** 1Department of Biomedical Engineering, University of Texas San Antonio, San Antonio, TX 78249, USA; sakethram.peri@utsa.edu (S.R.P.); delorenzo@uthscsa.edu (R.A.D.L.); 2Department of Emergency Medicine, University of Texas Health Science Center San Antonio, San Antonio, TX 78229, USA; 3Department of Mechanical Engineering, University of Texas San Antonio, San Antonio, TX 78249, USA; forhad.akhter@utsa.edu

**Keywords:** portable suction device, ISO 10079-1, portable suction device standards, airway clearance, standard developing organizations (SDO), aspirators, emergency medics

## Abstract

Airway clearance refers to the clearing of any airway blockage caused due to foreign objects such as mud, gravel, and biomaterials such as blood, vomit, or teeth fragments using the technology of choice, portable suction devices. Currently available devices are either too heavy and bulky to be carried, or insufficiently powered to be useful despite being in accordance with the ISO 10079-1 standards. When applied to portable suction, the design and testing standards lack clinical relevancy, which is evidenced by how available portable suction devices are sparingly used in pre-hospital situations. Lack of clinical relevancy despite being in accordance with design/manufacturing standards arise due to little if any collaboration between those developing clinical standards and the bodies that maintain design and manufacturing standards. An updated set of standards is required that accurately reflects evidence-based requirements and specifications, which should promote valid, rational, and relevant engineering designs and manufacturing standards in consideration of the unique scenarios facing prehospital casualty care. This paper aims to critically review the existing standards for portable suction devices and propose modifications based on the evidence and requirements, especially for civilian prehospital and combat casualty care situations.

## 1. Introduction

Airway obstruction is a blockage in any part of a patient’s airway and is often caused by salivary secretions, hemorrhage or the accumulation of vomitus, broken teeth, bone fragments and other debris that prevent natural air circulation [1,2,3,4]. Hence, clearing out the blockage is the initial procedure to revive the patient in any prehospital scenario and is enshrined in the resuscitation mnemonic “ABC”—Airway, Breathing and Circulation [5,6,7]. Airway obstruction is a major priority in prehospital emergency care and is also the second leading cause of preventable battlefield death [1,8,9,10]. In situations where a medical vacuum inlet is not available, such as during intrahospital transfers or in the pre-hospital environment, portable suction units are the technology of choice [11,12]. Suction devices, also termed aspirators, are powered or unpowered devices employed to remove infectious materials from wounds or fluids from a patient’s airway or respiratory support system [13,14]. The Association of Anesthetists of Great Britain and Ireland (AAGBI) lists portable suction among the essential equipment for pre-hospital anesthesia [12]. In this paper, we will preferentially use the term suction. First responders, including both civilian emergency medical services (EMS) and combat medics, are trained in using both manually operated and powered portable suction devices of various specifications available on the market [15,16,17].

Unfortunately, many of these commercial portable suction devices have not been scientifically validated for key performance measures relevant to prehospital care, let alone tactical combat casualty care [1,17,18]. Feedback from first responders, both civilian and military, described the major deficiencies of current commercial devices is that they are either too large and heavy to carry to the scene of an emergency (for powered systems) or too weak to effectively move viscous liquids (for manually operated systems) [1,19,20]. As combat medics are allowed discretion in which devices to include in their carry kits, this leads to elective exclusion of portable suction devices [21]. Discussions with civilian prehospital providers highlighted a similar reluctance to carry suction devices if they must walk a significant distance from their ambulance [1,22]. The omission of a suction kit leads to poor visualization of an injured airway which directly contributes to the need to employ a surgical procedure that is less optimal and fraught with complications [1,8,23]. Thus, patients suffer a tragedy of omission, as they are deprived of access to lifesaving technology due to the available devices not being compatible with end-user needs [24,25].

Figure 1 show some of the commercially available manually operated or battery powered suction units specifically advertised for use in prehospital tactical environments [20]. Most suction devices advertised for use in prehospital tactical environments were well within the specifications dictated by the ISO 10079-1 standards, yet no device on the market meets even the most basic field requirements on size, weight, and suction performance [1,19]. “Standards are the building blocks that provide for repeatable processes and composition of different technological solutions to achieve a robust end result.” [26]. An updated set of standards is required that accurately reflects evidence-based requirements and specifications, which should promote valid, rational, and relevant engineering designs and manufacturing standards in consideration of the unique scenarios facing prehospital casualty care.

The aim of this paper is to provide a primer on the standards ecosystem for medical devices with an eye towards reviewing the existing standards for portable suction devices. We also propose modifications based on the evidence and requirements, especially for civilian prehospital and combat casualty care situations.

## 2. The Standards Ecosystem

Standards developing organizations (SDOs) provide a platform for stakeholder collaborators to promote standards that promulgate the safety and efficacy employing medical devices while supporting compliance with the various regulatory frameworks around the world. Today, at least 1102 standards have been developed by different governing and consulting bodies to guide the design and manufacturing of medical devices [30].

Table 1 lists out the common SDOs (manufacturing and clinical) and regulatory agencies that have produced standards or guidelines relevant to medical devices. Medical device standards facilitate communication between stakeholders across domain borders, borders of the manufacturing system hierarchy, and between lifecycle phases [26]. During the lifecycle of a medical device product, the standards facilitate development of performance metrics, characterization and testing methodologies, manufacturing practices, product standards, scientific protocols, compliance criteria, ingredient specifications, and labeling, among other technical and policy criteria. Compliance with manufacturing and clinical SDO standards is important during the first three stages of the product life cycle: design, testing, and production. Compliance with regulatory agency standards plays a key role in the fourth and final stage: regulation. In an ideal situation, the manufacturing and clinical SDOs collaborate to develop a recommended set of standards, which would be enforced by a regulatory agency as in Figure 2. In reality, collaboration between clinical and manufacturing SDOs is rare and appears to be completely lacking for portable suction devices [31].

Gaps in product design and performance arise due to the scarcity of collaboration between manufacturing and clinical SDOs. In the case of portable suction devices, the primary manufacturing SDO (International Organization for Standardization, ISO) provides primary performance standards that do not appear to be clinically relevant. For example, suction device standards specify a minimum air flow rate when their primary use is to evacuate viscous liquids. In a related fashion, clinical SDOs have not coalesced around the specific airway clearance needs of patients and provide only circumspect guidance. Enhancing collaborations between manufacturing and clinical SDOs would lead to increased shared expertise across these often artificial boundaries with definite benefits to the end users and patients [31].

## 3. Standards Pertaining to Portable Suction Devices

### 3.1. Manufacturing Standards

ISO 10079 provides the standards for medical suction devices. Most of this ISO standard describes good manufacturing practices, safety standards, and design implications presented in format and verbiage in a style easily accessible to clinicians. Though ISO 10079 specifies safety and performance requirements for powered medical and surgical suction equipment, it does not specify the design and performance requirements for portable suction devices in prehospital situations [1]. Compliance with ISO standards are voluntary [32]. A few governmental bodies, such as the Food and Drug Administration (FDA), mandate compliance to a few ISO standards [1,13,20]. The design requirements described in ISO 10079 provide manufacturers with the mandatory features the design must include, helping to ensure medical suction equipment meets user requirements. In addition, these requirements detail safety features the device must include to mitigate the risk of device failures. Overall, ISO 10079 makes attempts to define a set of characteristics that powered medical devices must meet to fulfill the expectations of clinicians and patients. However, it is lacking in accurately reflecting current clinical needs for portable airway clearance systems [1].

### 3.2. Clinical Standards

Few clinical SDOs have made commentary regarding portable suction standards, and the few that have only provided general guidance. For example, the Committee on Tactical Combat Casualty Care (CoTCCC) specifies that airway management is of critical importance, as it is the second leading cause of preventable battlefield deaths [33,34]. CoTCCC guidelines for managing airway blockage include a series of physical maneuvers wherein to maintain an open airway for removing blood, mucus or vomit, the casualty is placed in the recovery position (laying on one side of the body, i.e., right lateral recumbent or left lateral recumbent) [33]. That the CoTCCC guidelines emphasize maneuvering the patient to recovery position and less on powered portable suction devices reflects the well-known reality that medics rarely carried such systems. In any event, CoTCCC guidelines do not specify particulars such as tubing and catheter sizing when evidence is available to show the superiority of specific combinations [9,10]. In the absence of other organized relevant clinical SDOs, players in the field with a stake in portable suction devices must turn to the best available evidence in the literature [1,4]. Unfortunately, there is a dearth of such information available. An approach that does not require the direct input of SDOs but instead queries end-users directly has been demonstrated for other airway devices [35].

To advance the state of the art, we propose the following specifications (Table 2) as a refinement to existing standards. These suggestions are based on a synthesis of the best available evidence in the literature, with the following paragraphs highlighting the derivation of some of the individual recommendations.

We further propose that manufacturing and clinical SDOs formally collaborate to review current suction device standards and propose future standards to better meet end-user requirements. In general, petitioning the SDOs is the first step towards proposing a revision to the current standards. The ISO committee holds meetings to review current standards and the current state of the art in technology. The committee also gathers input from their members and stakeholders to determine if revisions are needed to standards, which can be implemented through a system of voting.

## 4. Rationale for the Proposed Standards/Guidelines

### 4.1. Dimensional

An extensive survey of first responders suggests that powered suction devices are simply too heavy to be carried in the combat medic’s aid kit [1,19,20]. In comparison with battery operated suction devices, manual suction devices typically weigh far less but are unable to provide effective, reliable suction during emergency or combat situations. ISO 10079-1 states that the mass of the portable or handheld suction device shall not exceed six (6) kilograms [32]. In a study conducted on the inventory of 44 combat medics kits, only 15% of medics carried any suction device and all were lightweight manually operated disposable units [21]. Evaluation of the 23 currently available portable suction devices by De Lorenzo et al. demonstrated that the average weight of existing powered suction devices on the market is 3.69 kg with a range of 1.18–7.03 kg [1]. For manual suction devices, the average is 0.3 kg with a range of 0.19–1 kg. Though most of the powered devices align with ISO 10079-1 standards, these are not used by combat medics due to size and weight constraints. As the manually operated portable suction devices are not preferred by medics due to ineffective functionality, this is a clear sign that the current standards lack clinical relevancy.

The combat medic’s kit routinely weighs in excess of 45 kg [8], meaning that the average powered suction device on the market (approximately 3.6 kg) would be approximately 8% of the kit weight. As these are considered too heavy to be included, standards should describe a lower limit. We propose to decrease the weight standard for suction devices to be considered “portable” to be <2.25 kg, corresponding to <5% of the kit weight. This limit should also be more than amenable to civilian prehospital providers, who are not typically as weight constrained as military medics.

### 4.2. Performance

The current ISO 10079-1 standard for primary performance is for maximum free air flow rate of 20 L/min, which is confusing given the intended use of the device is to suction viscous liquids. While air is a fluid, there are many reasons that this number is misleading. First of all, specification of the “maximum free air flow rate” allows manually operated unit manufacturers to compress their units with forces beyond human capability to create a single instant of highly rapid air flow, allowing claims of similar performance to powered units on this metric [39]. Furthermore, the flow of air from the vacuum connector may not guarantee that there will be adequate negative pressure and flow to remove liquids through the comparatively high resistance suction tubing and catheter [39]. An experiment by Paulsen et al. shows that air flow rate at the tip of suction tubing was 47.6 L/min; for the same pressure, the water flow rate was measured to be 3.93 L/min [39]. As the flowrate of a fluid through a pipe is inversely proportional with viscosity as specified by the Hagen–Poiseuille equation (assuming laminar flow), this means that flowrates fall dramatically when evacuating viscous liquids. While this is arguably obvious, we suggest it should be specified within the relevant standards.

Device liquid evacuation performance should be measured using a fluid that mimics the viscous secretions and blood anticipated in the prehospital environment rather than the current standard of maximum flowrate of air. Liquid flowrate for a viscous liquid acts as a better standard as it is more appropriate for the intended usage of the device. ISO mentions specific gravity and recipe for simulated vomitus solution but do not mention the required liquid flowrate to suction the simulated vomitus liquid. Hence, we suggest updating the standards for better relevancy as they should provide a testing liquid with a specific viscosity as well as a required average fluid flow rate.

### 4.3. Testing

Interestingly, ISO 10079 provides a recipe for a vomitus fluid mimic that specifies mixing Xanthan gum and 1 mm glass beads in water. This mimic solution is intended to simulate the blood, emesis, human teeth, and bone fragments potentially requiring evacuation during an intervention following trauma, and goes so far as to specify a viscosity of 22.71 cP [8]. This implies that the rapid evacuation of viscous fluids to save a patient’s life is anticipated by the ISO. However, the standards do not specify a desired performance metric, such as evacuation flowrate, for suction devices being tested with this mimic solution. Furthermore, studies show that the viscosity of various substances that may require suctioning during anesthesia can have viscosities up to 230 cP [1,39]. Another departure from the clinical case is that broken bones, human teeth, and the contents of human emesis typically have particles larger than 1 mm in characteristic diameter. For example, the average characteristic length of an adult human molar teeth is approximately 8–9 mm with a weight ranging from 3–5 gm [8,40]. A study by Akhter et al. demonstrated that the capabilities of a suction system to intake solid particles and resist line clogging is highly dependent on particle geometry and mass [8]. While it is difficult to predict the size of semi-solid food particles from emesis or bone fragments from traumatic damage, it is reasonable to assume that 1 mm spheroids do not properly capture the clinical case. Therefore, we propose that the ISO standard could be strengthened by incorporating a variety of solid particles that would mimic the size and weight of human teeth. It is further recommended to employ a sufficiently large suction tube and catheter with the portable suction unit to accommodate such particulates.

### 4.4. Design

Large-bore tubing is preferred to smaller tubing, as hydraulic resistance is inversely proportional to the fourth power of the tubing’s radius [37,39,41]. Interviews with combat medics on required characteristics indicated the interior diameter of the tubing is a priority for effective suction [8]. ISO 10079-1 states that suction tubing must have an inside diameter of no less than 6 mm. In a study conducted by Vanderberg et al., the 6 mm tubing used became obstructed in 11 out of 12 instances while evacuating simulated vomitus solution [36,37]. Though the study was conducted using wall suction available in hospitals, the vacuum pressure employed was 500 mmHg, which is above the ISO standard for minimum requirements for vacuum pressure, suggesting portable units could fare even worse. A prototype developed by Akhter et al. demonstrated that tubing with an internal diameter of 10 mm could provide effective suction when paired with larger diameter off-the-shelf suction catheters (e.g., Wide Yank, 11 mm ID) [8]. We propose to increase the minimum standard for tubing Internal Diameter (ID) from 6 mm to 8 mm, corresponding to a 68.4% decrease in hydraulic resistance, for reducing clogging and obstruction during performing suction.

The suction catheter is an essential component of portable suction devices. The catheter is inserted into the oral cavity to clear the airway by evacuating any matter, liquid or solid, obstructing air passage to the lungs. There are several varieties of suction catheters currently available on the market that are compatible with portable suction units. ISO 10079 does not specify the design criteria of a suction catheter to use in a powered portable unit, and instead regulatory bodies such as the FDA consider it a separate device [13,32].

A comprehensive market survey revealed that most portable suction systems do not specify a preferred suction catheter. Given the influence of catheters on suction device performance, it is important to identify the best performing suction catheters for prehospital civilian and military use. Prior studies by this group compared five popular, commercially available suction catheters: Yankauer, Wide Yank, Poole, Argyle, and Flange type catheters [8]. The volumetric flow rate of viscous vomitus and blood mimicking solutions were evaluated at a fixed vacuum pressure of 650 mmHg with each catheter. Additional experiments sought to characterize the suction catheter clog frequency and average uninterrupted operation time between consecutive clogs as the catheters evacuated a volume of cream of mushroom soup.

Experimental results demonstrated that the Wide Yank suction catheter (11 mm inlet diameter) had a higher flow rate and less clog frequency than the current clinical standard Yankauer tip (4 mm inlet diameter). Another recent study on suction catheter strengths studied the fact that a large-bore catheter had superior suction rates compared with the Yankauer, despite fluid viscosity [41]. As the suction catheter is intended to evacuate viscous solutions with solid particles, a higher inlet diameter without any sharp bending is preferred to improve the volumetric flow rate and less clogging.

Another important design feature of a suction catheter is the circumferential air vents at the tip of the catheter. The standard Yankauer tip has four air vents of 2.3 mm diameter each at the tip to release the vacuum when the main inlet is obstructed by soft tissue. This is a safety feature to protect soft tissue from high vacuum pressure when the tip makes contact. Prior studies showed that these air vents tend to clog frequently when solid particles are present, which are typical when a patient vomits or if the mouth has had a traumatic impact that can produce broken teeth and bone fragments. Hence, a suction catheter with larger air vents (2.5–3 mm diameter) at the tip is preferred for the portable suction units most likely to be initially employed for those having been in traumatic accidents.

Reliability and battery life are obvious, important performance characteristics for a portable device intended for prehospital use [1]. Requirements regarding battery power and lifespan testing are sufficiently provided in ISO 10079-1 (Table 2), describing to manufacturers the minimum metrics needed for effective suctioning. Due to the unpredictable frequency of clinical need, the potential for lack of maintenance and deficiencies in routine inspection may impact the functional status of these devices when they are most needed [42]. An inspection on 9631 EMS suction units demonstrated a failure of 233 units in which 54.1% (126 units) failed due to battery failure [43]. Therefore, we suggest a requirement for including an indicator for battery life and specified frequency of required inspection.

Open suctioning is the most-used method for clearing airway secretions during mechanical ventilation in the intensive care unit (ICU). Open suctioning is conducted on a patient by disconnecting the patient from the ventilator and introducing the suction catheter or tubing, while in closed suctioning, airway secretion is conducted without disconnecting the ventilator. In the case of emergency situation, performing closed suctioning is impractical with portable suction devices. As a result, the medic is left with the only option of performing open suctioning, which leads to a high risk of bacterial or viral contamination to the medic. The potential advantages of a closed system over the conventional open suction technique include a reduction in contamination with potentially infectious organisms and maintenance of ventilator parameters [42]. A study conducted by Yu H.-J et al. shows that air within 1–2 m of endotracheal intubation site is contaminated after open suctioning [44].

The current study by El-Atab, N. et al., shows that the size of SARS-CoV-2 virus particulates range from 65 nm to 140 nm and the transmission can be contained by a nanopores membrane of 5 nm pores [38]. We propose a new requirement of including hydrophobic viral filters with filter size less than 5 nm within the portable suction devices to reduce the risk of transmitting infectious diseases to healthcare workers and nearby patients.

## 5. Future Direction for Sensor Integration

Suctioning is one of the primary functions performed during a pre-hospital emergency [5,6,7]. Though suction devices have been on the market for more than 100 years, few employ modern feedback technology, such as sensors. This may signal an unrealized opportunity to integrate sensors to align performance of the device with the patients’ needs and end-user’s preferences. For example, biological sensing in the mechanical domain provides unique opportunities to measure forces, displacements, and mass changes from cellular and subcellular processes [45]. We identify several key areas in which the addition of sensor technology can improve performance and features of portable suction devices.

Continuous suction for more than 15 s leads to a risk of hypoxia for a patient. The integration of an oxygen sensor at the tip of the catheter can alert the operator when the FiO_2_ is below an unsafe threshold; the process could be automated to increase supplemental oxygen flow rates to compensate [46]. Continuous suction (for more than 15 s) can also lead to necrosis which leads to tissue damage [47]. Risk of necrosis can be mitigated by incorporating sensors to detect pressure and possibly local tissue oxygen saturation. The feedback produced by the sensor could automatically adjust vacuum pressure (and thus flowrate) to mitigate harmful effects.

The Hagen–Poiseuille equation states that viscosity is inversely proportional to the flow rate of the fluid [36,37]. As flowrate increases with a decrease in viscosity, the incorporation of a viscosity biosensor can alter the device’s performance by changing the vacuum pressure (and thus flowrate). A biosensor can protect providers as well as patients: pathogen sensors could detect and trigger mechanisms to restrict contaminated aerosols spewing out of the suction devices. Other sensors that can potentially be integrated into portable suction devices are shown in Table 3.

Technological development of wearable sensors and systems has reached the point of being ready for clinical applications [48]. Synchronizing wearable sensor capabilities to portable suction devices could potentially enable the caregiver to continuously monitor the patient. A recent group demonstrated a wearable wireless device based on near infrared spectroscopy that could measure tissue oxygen saturation [49]. The integration of such a system could enable the monitoring of tissue oxygen saturation to help identify whether overaggressive suction is blanching tissue or if additional injury is present.

None of the current commercial portable suction devices employ a sensor to actively monitor device performance [19]. In addition, most commercially available portable suction devices specify changing and disposal of the canister between uses, but do not include the pump and primary housing. A recent study of oral suction used in dentistry has shown that non-disposable suction system components are prone to interpatient contamination [50]. Hence, the suction units are disinfected daily to avoid cross contamination [50]. Similarly, portable suction devices are prone to internal contamination and aerosol generation. Nanomechanical biosensors have become a promising technology to detect bacterial contamination with increased effectiveness and reduced footprint [51]. Inclusion of such nanomechanical biosensors into portable suction devices may aid in alerting users to potential contamination.

## 6. Conclusions

Airway management is crucial to the survival of patients with major airway trauma in first responder situations. Clearing airway obstruction is so important it is codified in the prime ABC mnemonic of resuscitation—Airway, Breathing and Circulation. On the battlefield, airway obstruction is the second leading cause of preventable deaths due to inadequate airway management devices. Currently available devices are either too heavy and bulky to be carried, or insufficiently powered to be useful despite being in accordance with the ISO 10079-1 standards. Standards laid down by manufacturing SDOs, clinical SDOs and regulatory agencies act as guidelines to manufacturers in developing an effective suction device. Due to the under-development and lack of meaningful collaboration between manufacturing SDOs and clinical SDOs, most marketed suction devices do not meet even the most basic field requirements on size, weight, and suction performance.

Standards pertaining to portability, flowrate, viscosity, vomitus solution, tubing internal diameter, suction tip, battery power, and viral filter are suggested to meet the requirements of the end user, civilian emergency medical services (EMS) and medics. Below is the list of recommendations the authors propose as updates to current ISO 10079-1 standards.


Weight of the suction devices shall be <2.25 kg rather than <6 kg to be considered portable, which leads to suction devices occupying 5% of the weight in the medical bag rather than 8%.“Maximum air flow rate” is not a commendable specification to standardize the suctions device as they are used to draw fluid (blood, secretions, bone fragment and teeth fragment) and not air in pre-hospital situations.ISO 10079-1 recommends a simulated vomitus solution to characterize the suction device but does not provide neither the required liquid flow rate to draw the simulated liquid nor the required viscosity of the solution.Large-bore tubing is preferred to smaller tubing, as the hydraulic resistance is inversely proportional to the fourth power of the tubing’s radius. Increasing the minimum standard for tubing Internal Diameter (ID) from 6 mm to 8 mm, corresponding to a 68.4% decrease in hydraulic resistance, for reducing clogging and obstruction during performing suction.Mandatory inclusion of suction catheter to the suction device for improving the volumetric flow rate and minimizing the clogging.Inclusion of battery life indicators and specified frequency of required inspection to minimize the failure of device during its use.First responders are at high risk of being exposed aerosolized viruses. The inclusion of viral filters to the exhaust of the suction devices minimizes their risk.


## Figures and Tables

**Figure 1 sensors-22-02515-f001:**
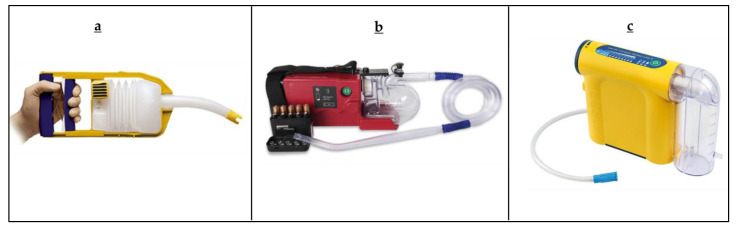
Commercially available portable suction devices, (**a**) Manually operated portable suction device—Laerdal VVAC [27]; (**b**) Battery operated portable suction device—SSCOR Quickdraw [28]; (**c**) Battery operated suction device—Laerdal LCSU 4 [29]. Reprinted with permission to reprint images of the portable suction devices were obtained from respective manufacturers.

**Figure 2 sensors-22-02515-f002:**
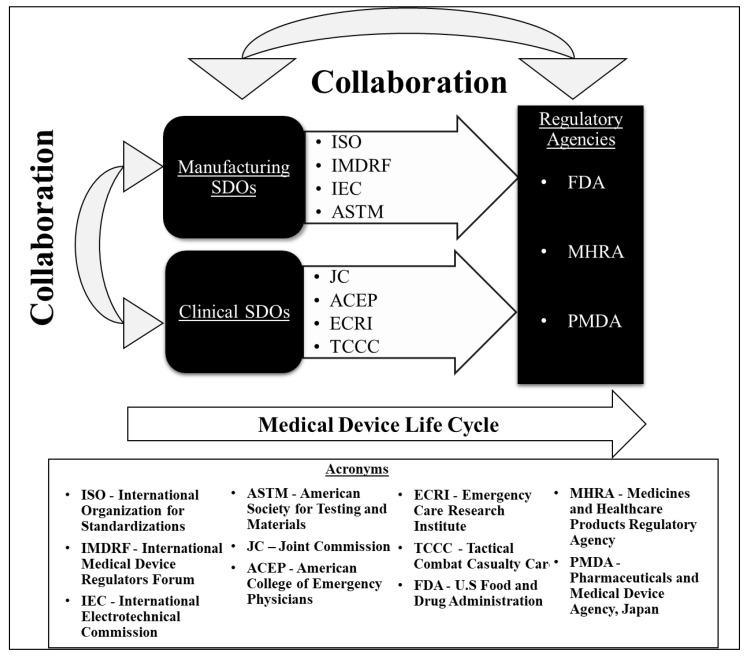
Ecosystem of Standards. Reprinted with permission Idealized depiction of the inter-relationships informing medical device standards. The organizations depicted are examples relevant to suction devices and will vary with the device examined.

**Table 1 sensors-22-02515-t001:** Standards developing organizations (SDOs) with examples relevant to suction devices.

Manufacturing SDOs		International Organization for Standardization (ISO) International Medical Device Regulators Forum (IMDRF) International Electrotechnical Commission (IEC) American Society for Testing and Materials (ASTM)
Clinical SDOs	Accrediting Bodies	Joint Commission (JC)
	Medical Societies	American College of Emergency Physicians (ACEP)
	Private, Non-Profit Organization	Emergency Care Research Institute (ECRI)
	Governmental, Nonregulatory Entities	Committee on Tactical Combat Casualty Care (CoTCCC)
Regulatory Agencies		U.S Food and Drug Administration (FDA) Medicines and Healthcare Products Regulatory Agency, U.K. (MHRA) Pharmaceuticals and Medical Device Agency, Japan (PMDA)

**Table 2 sensors-22-02515-t002:** New guidelines proposal for powered portable suction device.

Type of Characteristic	Requirement	Existing Standards/Guidelines from ISO 10079-1	Proposed Standard/Guidelines
Dimensional	Weight	<6 kg	<2.25 kg
	Size (Length × Width × Height)	60 × 30 cm (with no mention of height)	30 × 10 × 10 cm
Performance	Flow Rate	>20 L/min free air flow	>1 L/min of water-based viscous solution
	Viscosity	It only provides a recipe for vomitous solution but not the viscosity	20 to 25 cP
	Battery Power	operate > 20 min @ free air flowrate > 20 L/min and a vacuum > 40 kPa	Indicators for, Battery Life
	Low Vacuum Pressure Device	>20 kPa (150 mmHg)	Existing standard is reasonable
	High Vacuum Pressure Device	>60 kPa (450 mmHg)	
Design	Tubing Diameter	>6 mm	>8 mm [36,37]
	Suction Nozzle	Not stated in ISO or elsewhere	Yankauer Suction Catheter or equivalent [36,37]
	Viral/Bacterial Filter	Not stated in ISO or elsewhere	Hydrophobic Viral Filter (filter size < 5 nm [38])
Testing	Vomitus Solution	10 g Xanthan Gum, 1 L distilled water, 1 mm glass beads	10 g Xanthan Gum, 1 L distilled water, 7–10 mm & 3–5 g human teeth mimic
	Clog Resistance	Not mentioned	Clogging standard proposed by Akhter et al. [2]
	ASTM Testing/CCG Testing	Not mentioned	Need to evaluate device performance after transit simulation

**Table 3 sensors-22-02515-t003:** Potential sensor applications for portable medical suction devices.

Biosensor	Application	Clinical Effect
Tissue pressure	Avoid excess catheter tip pressures	Prevent local tissue necrosis
Oxygen saturation	Avoid inadequate FiO_2_	Prevent hypoxemia
Tissue viability (e.g., tissue blood flow or oxygen saturation)	Differentiate necrotic from viable tissue	Clearing necrotic tissue debris while avoiding viable tissue.
Ultrasound, video, or other imaging modality	Identify anatomic structures	Guide and maintain catheter tip positioning inside body cavities
Fluid viscosity/particulate content	Detect changing body fluid viscosities	Enables machine to optimize performance (e.g., vacuum pressure)
Hematocrit/hemoglobin mass	Measure proportion of red cell mass in evacuated fluid	Real-time measurement of red cell mass during cell saver operations
Pathogens (e.g., bacterial, or viral)	Avoidance of cross-contamination and exhausting of pathogens into local environment	Protect providers from contaminated aerosols

## Data Availability

Not applicable.

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
