# Peer review of "Portable Medical Suction and Aspirator Devices: Are the Design and Performance Standards Relevant?"

_sensors, 2022, doi:10.3390/s22072515_

Round 1

Reviewer 1 Report

1. In this abstract, the background information is too long, and the innovations and characteristics of this paper should be strengthened.

2. Recently, wearable sensors and devices are well developed, which can be discussed in portable medical devices.

Author Response

Authors appreciate the valuable suggestions from the reviewer. As per the reviewer’s feedback, changes were made. Please see attached

Reviewer 2 Report

The authors review existing standards for the Portable Medical Suction and Aspirator Devices and propose new standards to improve performance of these devices.

Generally, the topic is explained well in the manuscript. The authors can clarify or explain how the implementation of the proposed modifications in manufacturing and clinical sides would be. What would be the time frame to implement the proposed standards?

Also, are there any sensors to sensorize Portable Medical Suction and Aspirator Devices in the market. A few examples may help to explain the current sensor technology.

Author Response

(The authors gave the same response as above.)

Reviewer 3 Report

  1. It would be great if the readers can see images of portable suction devices.
  2. Although ISO does not provide standard for liquid flow rate, Does it mean that it is not required?

Author Response

(The authors gave the same response as above.)

Round 2

Reviewer 2 Report

The authors respond to the reviewer’s comments sufficiently. The submitted study can be accepted as a publication.